# HierT2S: Enhancing Part-Level Text-to-Shape Generation via Hierarchical Structure Modeling

## Abstract

Text-driven 3D shape generation still faces key challenges, especially in achieving high levels of control over the generated outputs. Paticularly, existing text-to-shape methods ignore the explicit modeling of hierarchical structures in the text and 3D shapes, which makes it hard for using long text descriptions with multiple prompts to guide the coherent part-level 3D shape generation. In this work, we introduce HierT2S, a framework that integrates a hierarchical tree representation with a conditional diffusion model, to enhance the generation of 3D shapes with coherent structures induced by the hierarchical and structured text representations. The key idea is to first segment the input text into several clusters and construct a hierarchical tree representation, with each node representing a parent entity or the fine-level part components. Then, we process the lower-level clusters of the tree with a relation graph module which uses self-attention mechanism to aggregate the relationships of the clusters, and generate a new sequence containing the processed text features. Finally, the text features are embedded into the 3D feature space and used for learning the 3D shape generation by a conditional diffusion model, where the sparsely implicit parsed hierarchical tree graph further enhances the structural details of the generated 3D shapes, leading to results that are close to structure-aware generation. We conducted comprehensive experiments on the existing text-to-shape pairing dataset Text2Shape, and the results demonstrate that our model significantly outperforms current state-of-the-art methods. Moreover, our method can enable progressive part-level 3D shape manipulation and modification guided by the partially modified text prompt.

## 1 Introduction

The text-to-3D shape generation field has advanced notably with the use of generative models like diffusion models (Ho et al., 2020), enabling the creation of high-quality and detailed shapes (Chen et al., 2024b; Chu et al., 2024; Liu et al., 2024b; Cheng et al., 2023). While text-based input provides a flexible and intuitive way for guiding 3D shape generation, accurately producing geometries that match the text remains challenging, especially for objects with hierarchical structures like chairs and tables. Since both text and 3D shapes have inherent hierarchical features, we believe it is critical to capture the relations between the hierarchical structures of those two modalities to ensure consistent text-to-3D generation.

Human naturally use hierarchical reasoning to perceive and represent complex information about objects' appearance, structure, and function (Baillargeon, 1996). This intuitive ability allows human to seamlessly integrate global and local dependencies within both visual and linguistic patterns (Biederman, 1987). For intelligent algorithms, however, understanding the structural connections between text and 3D shapes is a substantial challenge. For example, a detailed prompt such as "a solid locking chair made of grey tiles with a curved back and wheeled legs" may cause issues for a system which only utilizes global-level priors from the training data. Specifically, some high-frequent co-occurring patterns such as "straight legs" may affect the learning and generation of "wheeled legs" specified in the given text. This example underscores the importance of understanding and leveraging the hierarchical structures of both text and shapes to enhance the fidelity and specificity of generated 3D shapes. While existing works (Achlioptas et al., 2018; Xu et al., 2019) seek to learn

Figure 1: Overview of our pipeline. A Hierarchical Tree $\mathcal{G}$ is constructed based on the text input, which corresponds to the structure of 3D shapes. Then, the features extracted from the lower-level nodes of the tree are processed and used a conditional diffusion model for structure-aware generation and modification.

cross-modal mappings directly from text-3D pairs or utilize pre-trained text-to-image models (Li et al., 2024; Zhou et al., 2022; Zhang et al., 2023) to guide 3D modeling, these approaches still fail to address the hierarchical structure in both 3D shapes and natural language.

Motivated by reasoning ability of the hierarchical structures inherent in human intelligence, we propose HierT2S, a novel framework that exploits these structures in text-to-shape generation to achieve stronger semantic consistency and structure-awareness. Different from existing methods that typically learn direct mappings between text and 3D data, our approach captures the joint hierarchical dependencies of these modalities. In detail, the input sequence is first parsed into a hierarchical tree and then softly segmented into several clusters using a probabilistic graphical model based on the attention mechanism, capturing the leaf nodes of the internal entities. Then, we train a conditional diffusion model using the latent features of the clusters in the lower layers of the hierarchical structure of the segmented new sequence, achieving structure-aware text-to-shape generation.

Our approach has been evaluated on the Text2Shape (Chen et al., 2019) dataset, demonstrating significant improvements in generation quality while preserving the hierarchical characteristics. Furthermore, benefiting from structure-awareness ability, our approach enables structure-aware text-guided 3D shape manipulation and progressive modification. In summary, our contributions are as follows:

- By explicitly modeling hierarchical structure in text-to-shape generation, our proposed HierT2S achieves stronger semantic consistency and structure-awareness, enabling structure-aware manipulation and progressive modification.

- The proposed relation graph module effectively captures the hierarchical relationship between text and 3D shapes without requiring direct supervision on 3D part-level annotations, relying solely on general text-to-3D pairs.

- Comprehensive experiments on the Text2Shape dataset demonstrate the effectiveness of our method, with substantial improvements attained over existing approaches in generation quality and hierarchical structure preservation.

## 2 RELATED WORK

### 2.1 TEXT-GUIDED 3D SHAPE GENERATION

Recent advances in deep learning have revolutionized text-guided 3D shape generation, leading to various methods designed to produce 3D shapes and scenes based on text input. Early efforts primarily focused on modeling joint text-shape embedding spaces. However, generating high-quality 3D shapes directly from text remains challenging, primarily due to the difficulties in aligning textual descriptions with shapes and the scarcity of well-annotated paired datasets. The Text2Shape dataset (Chen et al., 2019) was a pioneering effort in this area, employing a GAN-based approach for shape

generation. Nonetheless, this method encountered limitations regarding resolution, quality, and cross-modal consistency. To address these challenges, subsequent methods have incorporated advanced techniques. For instance, recent approaches have leveraged pre-trained models such as CLIP (Abdelreheem et al., 2022b) and diffusion-based strategies (Abdelreheem et al., 2022a) to enhance the fidelity, visual realism, and structural accuracy of generated shapes. Additional works, such as (Li et al., 2023b; Cheng et al., 2022; Qian et al., 2024), utilize discrete autoencoders to capture block-based shape priors, which are processed by transformers for autoregressive shape generation. More recent diffusion-based methods (Zhao et al., 2024; Li et al., 2023a; Cheng et al., 2023) further improve this framework by generating latent features that align more closely with VQ-VAE embeddings, resulting in shapes of higher accuracy and fidelity.

Despite these advancements, most previous studies have treated shapes as unified entities, primarily relying on text representations that focus on linguistic features (such as sentences or words). This approach often fails to effectively convert sentences containing multiple descriptors into 3D shapes with complex structural details due to trivial attention mechanisms. Our research promotes better enhancement of 3D shapes by transforming text into structure-awareness.

## 2.2 3D Structure-Aware Representation

Structure-aware representation encompasses techniques that capture and utilize hierarchical and relational information within 3D shapes, facilitating more nuanced and accurate shape generation. This approach is essential for effectively decomposing complex shapes into their constituent parts and understanding their geometric relationships. Recent research has introduced several methods for learning structure-aware 3D shape representations. For instance, in supervised learning, some methods (Chen et al., 2022; Mou et al., 2024) advocate using symmetry hierarchies to represent hierarchical shape structures . Recent progress has also been made in semantic-based shape decomposition, with methods like those presented in (Cai et al., 2022; Yang et al., 2024b) using learned operations to identify grammar-level shape components. In the realm of unsupervised learning, recent studies such as (Ouasfi & Boukhayma, 2024; Liu et al., 2024a; Lee et al., 2024) utilize implicit neural representations as a framework to capture complex data modalities, preserving structured features through enhanced boundary sampling and stabilization of the optimization process. Alternatively, DAE-Net (Chen et al., 2024a) employs a branched autoencoder to learn a set of deformable part templates and achieve part segmentation of shapes through affine transformations. However, most methods focus directly on the structural aspects of 3D shapes without considering how to further enhance 3D shape generation through the structure of text. Our approach aims to address this gap by introducing text structure awareness in the text-to-shape generation process.

## 2.3 Graph Network Guidance

Graph networks are essential for modeling intricate relationships and dependencies within data. Recent works (Yang et al., 2024a; Jiang et al., 2024) leverage graph priors to facilitate the transfer of commonalities and bridge the gap between visual and linguistic domains. Other studies (Wu et al., 2024; Huang et al., 2024) utilize scene graphs—composed of nodes and relationships—to analyze and interpret 3D scenes. Scene graphs, generated from textual descriptions, capture expressive structural relationships among entities, enhancing the alignment between textual inputs and graphical models through the application of graph networks. They have been successfully employed in various tasks, including text-image matching (Huang et al., 2024), image processing (Gu et al., 2024), and caption generation (Luo et al., 2024). Recognizing that both shapes and texts are composed of structural elements, we choose to incorporate textual graphs as supplementary guidance alongside the generation of structural parts.

## 3 Preliminaries

**3D Shape VQ-VAE.** Modeling 3D shapes is challenging due to their high dimensionality. To address this, we compress 3D shapes from ShapeNet (Chang et al., 2015) into a lower-dimensional latent space using a 3D VQ-VAE (Van Den Oord et al., 2017). The 3D VQ-VAE consists of an encoder $E_\phi$ that maps 3D shapes into latent vectors, and a decoder $D_\tau$ that reconstructs the original 3D shapes from these vectors. Specifically, for an input shape $X$, represented by a volumetric

Truncated-Signed Distance Field (T-SDF) with dimensions $X \in \mathbb{R}^{D \times D \times D}$, the encoding process is defined as: $z = E_\phi(X)$ where $z \in \mathbb{R}^{d \times d \times d}$ represents the lower-dimensional latent space, with $d < D$. The quantization step (VQ) maps $z$ to the nearest entry in a learned codebook $Z$, and the decoder reconstructs the shape as: $X' = D_\tau(\text{VQ}(z))$. The encoder, decoder, and codebook are trained jointly to minimize the reconstruction loss, commitment loss (which encourages encoder alignment with codebook entries), and the VQ objective to improve quantization.

**Forward Process of the Latent Diffusion Model.** The diffusion model operates on the lower-dimensional latent variable $z_0 = E_\phi(X)$, where the model learns to generate samples by reversing a noise addition process, as described in (Ho et al., 2020). In the forward process, starting with the clean latent representation $z_0$, Gaussian noise is incrementally added over a series of time steps to produce a sequence of latent variables $\{z_t\}_{t=1}^T$. At each step $t$, the latent variable $z_t$ is obtained by diffusing the previous state:

$$z_t = \sqrt{\alpha_t} z_{t-1} + \sqrt{1 - \alpha_t} \epsilon_t, \tag{1}$$

where $\alpha_t$ is the noise schedule controlling the amount of noise added at each step, and $\epsilon_t \sim \mathcal{N}(0, I)$ represents Gaussian noise sampled from a standard normal distribution. This forward process continues until a predefined number of steps $T$, resulting in a latent variable $z_T$ that approximates random Gaussian noise.

## 4 METHOD

### 4.1 OVERVIEW

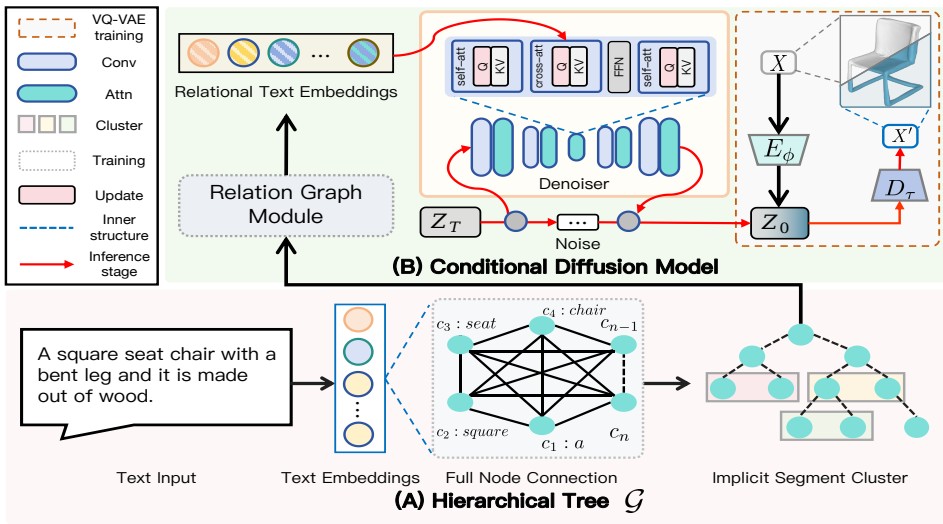

Figure 2: Method overview. HierT2S includes two phases: (A) preprocessing the text with the Hierarchical Tree $\mathcal{G}$, and (B) training the diffusion model's reverse process using local-level features of structural text entities preprocessed with hierarchical tree $\mathcal{G}$.

Figure 2 presents our framework, which enhances text-to-shape generation by integrating a semantic hierarchical structure. In the first stage (Figure 2 (A)), we introduce a Hierarchical Tree $\mathcal{G}$ to encode sentences containing multiple descriptive key prompts. We first segment the sequence into several clusters based on different parent entities, ensuring that the terms corresponding to lower-level identical components are retained within the same higher-level component. We then propose Relation Graph Module, a method that applies the attention mechanism to the integration of attention mechanisms into probabilistic graphical models, where the stacked attention layers effectively capture the relationships between entities within each cluster, subsequently performing top-down implicit parsing of the relevant internal components of these clusters. In the second stage (Figure 2 (B)), we integrate this tree-based hierarchical semantic structure into the conditional diffusion model. This

integration mitigates trivial global dependencies in long text descriptions, significantly enhancing the semantic capture of multiple prompts, especially those appearing later in the sequence. Our approach improves the capacity for 3D structural modeling, resulting in more expressive and diverse generations.

## 4.2 Hierarchical Tree of a Sentence

We draw inspiration from the Tree-Transformer (Wang et al., 2019) and recursively parse the input text sequence into a hierarchical tree from top to bottom. However, unlike the Tree-Transformer, which directly computes associations for all entities, we first calculate the entity correlation probability using a matrix $\boldsymbol{A} \in \mathbb{R}^{m \times m}$ to cluster entities by calculating association probabilities, where $m$ is the length of the entity sequence. This approach enables us to segment the sequence into several clusters based on different parent entities, thereby facilitating subsequent top-down implicit parsing of the relevant internal components of these clusters using attention layers.

Building on this hierarchical parsing strategy, we aim to ultimately align the semantic hierarchy with the entity hierarchy of 3D shapes, where segmented clusters represent parent clusters of entities. Figure 2 (A) contains a schematic representation of the process of dividing the fully connected node connections into implicit segment clusters, which involves clustering the lower-level entities that are associated with higher-level entities. For example, given a text input represented as $C = \{c_1, c_2, \ldots, c_N\} \in \mathbb{R}^{d \times N}$, each pair of nodes $c_i$ and $c_j$ is connected by an edge weighted by the attention coefficient $a_{i,j}$, and $c_1^1 = \{c_1^2, c_2^2\}$ indicates that the representation of the parent cluster at the first level, $\{chair\}$, consists of two child clusters at the second level, $\{seat, legs\}$. The subsequent layer provides descriptive terms for each child node, ensuring that terms belonging to the same constituent in a lower layer remain within the same constituent in higher layers.

Specifically, we use a $m \times m$ matrix $\boldsymbol{A}$ to compute the correlation between nodes, where $\boldsymbol{A}_{i,j}$ represents the weight of the clustering for node indices from $i$ to $j$, and the calculation of $\theta_n$ is similar to the Tree-transformer (Wang et al., 2019). By contrast, since we need to cluster entities, we use a hard segmentation approach to determine whether there is a connection between entities. This means nodes are clustered into the same group only when $\theta_n (x_t' = 1)$. This matrix $\boldsymbol{A}$ effectively captures the correlation probabilities between nodes $c_i$ and $c_j$, enabling direct modeling of relationships between any pair of nodes and providing a more flexible and explicit representation of their interactions. The detailed procedure is outlined in Algorithm 1.

**Algorithm 1** Segmentation Cluster Matrix $\boldsymbol{A}$

---

**Input:** $m \leftarrow$ size of matrix $\boldsymbol{A}$
Sequence of entities $C = \{c_1, c_2, \ldots, c_N\} \in \mathbb{R}^{d \times N}$
$\theta_n(x_n' = 1 | C)$: correlation probabilities for each entity $c_n$
**Output:** Matrix $\boldsymbol{A}$
**for** $i \leftarrow 1$ **to** $m$ **do**
    **for** $j \leftarrow 1$ **to** $m$ **do**
        **if** $i < j$ **then**
            Compute

$$\boldsymbol{A}_{i,j} = \prod_{t=i}^{j-1} \theta_t \left( x_t' = 1 \mid C \right)$$

        **else**
            $\boldsymbol{A}_{i,j} = \boldsymbol{A}_{j,i}$
    Set $\boldsymbol{A}_{i,i} = 1$
**return** $\boldsymbol{A}$

---

By evaluating the magnitude of $\boldsymbol{A}_{i,j}$, we determine the marginal probability of clustering $c_i$ with $c_j$, allowing the input sequence to be segmented into clusters. This approach enhances the decoding capabilities of conditional diffusion models for downstream tasks and effectively addresses the issue of neglected critical local context.

## 4.3 Relation Graph Module

We design a specialized PGM (Murphy, 2012) incorporating a self-attention mechanism (Vaswani, 2017), referred to as the Relation Graph Module, specifically tailored for Hierarchical Tree $\mathcal{G}$, further aggregating entity relationships within segmented clusters, thereby optimizing the model's likelihood. This module computes relation embeddings by leveraging the previously calculated correlation probabilities between nodes, enabling the model to encode relational information more effectively.

Generally, the factorization formula for a Markov Random Field (MRF) can be expressed as follows:

$$P(x_1, x_2, \ldots, x_n) = \frac{1}{\mathcal{Z}} \prod_{n=1}^{N-1} \psi_n(x_n), \tag{2}$$

where $\psi(\cdot)$ depends on the correlation coefficient of the cosine similarity between entity $c_i^d$ and $c_j^d$, and $\mathcal{Z}$ is the partition function, defined as:

$$\mathcal{Z} = \sum_{x_1, x_2, \ldots, x_n} \prod_{n=1}^{N-1} \psi_n(x_n). \tag{3}$$

As illustrated in Figure 3, we employ an attention mechanism to compute the correlation scores $\psi_n$ for each clique within the probabilistic graphical model, capturing the relationships between nodes, effectively modeling both local and global dependencies within the graph structure. Specifically, we process the input text $C$ to evaluate the relationship between an entity clusters $c_n$ and its two lower-layer child nodes $\{c_{n-1}, c_{n+1}\}$, which can clearly be achieved through the attention parameterization shown in the left of Figure 3.

The process of mapping entities into their respective key and query spaces is as follows: $K_{n-1} = E_K(c_{n-1})$, $Q_n = E_Q(c_n)$, and $K_{n+1} = E_K(c_{n+1})$, where $E_Q$ and $E_K$ are the embedding functions for queries and keys, respectively. The connection values $\{\lambda_{n-1}, \lambda_{n+1}\}$ for the entity $c_n$ are formulated as:

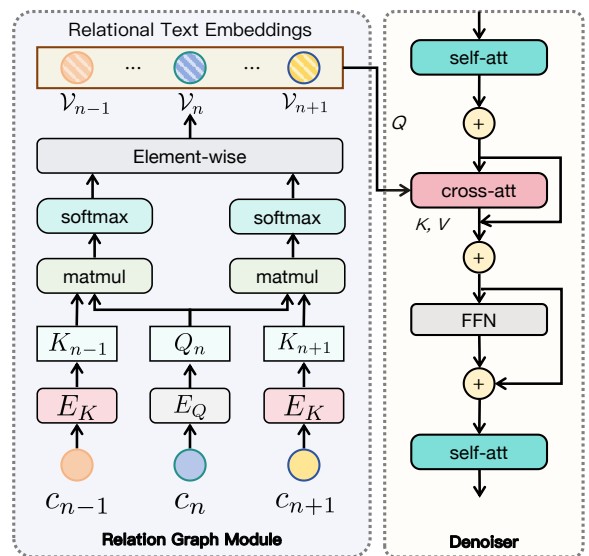

Figure 3: Left: the Relation Graph Module incorporates the attention mechanism to compute relation embeddings among entities. Right: relational text embeddings are incorporated as a condition into the denoiser of the diffusion model.

$$\lambda_n^{n-1} = \mathrm{softmax}(K_{n-1} \cdot Q_n); \quad \lambda_n^{n+1} = \mathrm{softmax}(K_{n+1} \cdot Q_n), \tag{4}$$

where $\boldsymbol{Q} \in \mathbb{R}^{N_Q \times d}$ and $\boldsymbol{K} \in \mathbb{R}^{N_K \times d}$ represent the matrices of queries and keys, respectively. The softmax function is utilized to normalize the attention scores. The potential function $\psi_n$ is determined by the similarity $\lambda_n^{n-1}$ and $\lambda_n^{n+1}$ between the current computation node and its neighboring nodes. We use the sigmoid function $\sigma(\cdot)$ (Rumelhart et al., 1986) to map the similarity values to the range $[0, 1]$ and control its sensitivity by setting the same breakpoint threshold as in the Tree-Transformer (Wang et al., 2019), formulated as:

$$\psi_n(x_n) = \sigma(\lambda_n^{n-1}) \cdot \sigma(\lambda_n^{n+1}). \tag{5}$$

Then we substitute Eq. (2) and Eq. (5) into Eq. (6), allowing us to compute the potential function of the entity:

$$P(x_1, x_2, .., x_{n-1} \mid C) = \frac{1}{\mathcal{Z}(x, C)} \prod_{n=1}^{N-1} \psi_n(x_n \mid C). \tag{6}$$

Finally, we use the partition clustering matrix $\boldsymbol{A}$ to update the node embeddings, thereby obtaining the semantic structure-aware features $\mathcal{V}$:

$$\mathcal{V} = (\boldsymbol{A} \otimes \mathrm{softmax}\left(\frac{\boldsymbol{Q}\boldsymbol{W}^Q \left(\boldsymbol{K}\boldsymbol{W}^K\right)^\top}{\sqrt{d}}\right))VW^V, \tag{7}$$

where $\otimes$ stands for the element-wise operation, $d$ is the dimension of $K$, resulting in the attention score matrix. In this way, we first reconstruct the original fully connected graph by constructing a sparse graph with several clusters, and then use an induced tree structure to effectively alleviate the issue of attention decay in long texts. As a result, in the task of 3D shape generation, prompts from any position in the sentence can be captured more effectively.

We propose to pre-train the Hierarchical Tree $\mathcal{G}$ by reconstructing the original node attributes. The goal is to let the node embeddings effectively capture and preserve the original attribute information, thereby enhancing the Relation Graph Module's ability to learn node features and improving its sensitivity to the original attributes. Specifically, we utilize the pre-trained prompt features from BERT (Devlin, 2018), denoted as $E_C$, as semantic anchors to capture the raw node information, where $y_i = E_C(c_i)$. For the relational text embedding $\mathcal{V}_i$, which captures the graph's structural information, we represent it using a multi-layer perceptron (MLP) as $\hat{y}_i = \text{MLP}(\mathcal{V}_i)$, and compare it with the raw attributes of node $y_i$. The loss function for reconstructing the original node attributes is defined as follows:

$$\mathcal{L}_{rec} = \frac{1}{|C|} \sum_{c_i \in C} \|\mathbf{y}_i - \hat{\mathbf{y}}_i\|_2 . \tag{8}$$

## 4.4 TRAINING

We train the entire network jointly in an end-to-end fashion to achieve high-quality 3D shape generation. Specifically, we have used a 3D-UNet based conditional diffusion model(Çiçek et al., 2016). Starting from random Gaussian noise $Z_T$ at time step $T$, the denoiser, utilizing and integrating structural relational text embeddings through a cross-attention mechanism, transforms the latent feature $z_t$ at time step $t$ to $z_{t-1}$. The training objective for the denoising process at each time step $t$ is to minimize:

$$\mathcal{L}_{CDM} = \mathbb{E}_{\mathbf{x}, \varepsilon \sim \mathcal{N}(0,1), t} \|\epsilon - \epsilon_\theta (z_t, t, v_i)\|^2 . \tag{9}$$

The complete training loss for the entire framework is a weighted sum of the reconstruction loss for the Hierarchical Tree $\mathcal{G}$ and the loss for the conditional diffusion model, with the weights to be $\lambda_1$ and $\lambda_2$ respectively:

$$\mathcal{L} = \lambda_1 \mathcal{L}_{rec} + \lambda_2 \mathcal{L}_{CDM}. \tag{10}$$

By jointly training the hierarchical tree and the diffusion model, the structurally aware text descriptions significantly enhance the denoising capability of the 3D U-Net. It enables the generated 3D shapes to achieve a higher degree of alignment with the input multi-prompt descriptions and also allow the model to achieve part-level high-fidelity 3D shape generation.

## 5 EXPERIMENTS

This section introduces our experimental design and implementation, followed by a comprehensive analysis of the results from various aspects.

**Settings.** To evaluate the performance of our method, we conducted a series of experiments on the paired text-to-shape dataset Text2Shape. First, we trained a 3D VQ-VAE on all the 3D shapes in the Text2Shape dataset (Chen et al., 2019), using the ShapeNet dataset (Chang et al., 2015). The 3D VQ-VAE compresses the T-SDF into compact latent features $Z$. This process involves converting the T-SDF into a mesh, followed by sampling 2048 points from each mesh. The dimensions of the latent features $Z \in \mathbb{R}^{16 \times 16 \times 16 \times 3}$. Then, we train a text graph node with the Hierarchical Tree $\mathcal{G}$. The number of channels for the text encoder is $d_h = 512$, and for inner layer features in the feed-forward network it is 2048. The Adam optimizer is used. The learning rate is initialized to $1 \times 10^{-5}$ and decays by 0.8 every 5 epochs, with 20,000 epoch. Next, we employ the 3D VQ-VAE decoder and trained a conditional diffusion model using the provided pretrain relational text embeddings. Our work utilizes the Adam optimizer to train the DDPM sampler for 200 steps with an initial learning

rate of $1 \times 10^{-4}$. During the shape modification phase, we fine-tune the model for 500 epochs using the same learning rate.

**Evaluation Metrics.** (1) CLIP-S: Following 3DQD (Li et al., 2023a), we use CLIP-S, which computes the maximum cosine similarity between $N = 9$ generated shapes and their text prompts. Each shape is rendered into 20 2D images from different views. During testing, we use a pre-trained CLIP model as the text encoder; (2) Intersection over Union (IoU): it measures overlap between generated and ground truth shapes; (3) Total Mutual Difference (TMD): it sums up pairwise differences among $N = 10$ generated shapes; (4) Earth Mover Distance (EMD): it measures the cost to transform one distribution into another.

### 5.1 TEXT-GUIDED 3D SHAPE GENERATION

We have compared with recent state-of-the-art approaches for text-guided 3D shape generation, including AutoSDF (Mittal et al., 2022), Shape-IMLE (Liu et al., 2022), SDFusion (Cheng et al., 2023), and 3DQD (Li et al., 2023a). While some of these methods can generate 3D shapes using various conditioning inputs, our evaluation focuses exclusively on the text-to-3D representation task, where text prompts serve as the sole conditioning input.

As shown in Figure 4, the existing methods face significant challenges in generating 3D shapes with high fidelity and structured details. For instance, with the text prompt "crisscross legs", both AutoSDF and Shape-IMLE struggle to generate precise structural details, while SDFusion has difficulty maintaining adherence to shape specifications. Additionally, some keywords placed at the end of a sentence (e.g., "back and head support") fail to fully capture the semantic information during the generation process. In contrast, our approach clearly demonstrates superior performance in generating high-quality 3D shapes with well-defined and coherent structural details.

For quantitative evaluation, we adopt IoU, CLIP-S, TMD and EMD to evaluate the generative quality and diversity of shape, respectively. As illustrated in Table 1, our model consistently surpasses existing methods across all metrics. These results indicate that our method effectively learns and utilizes structural text features.

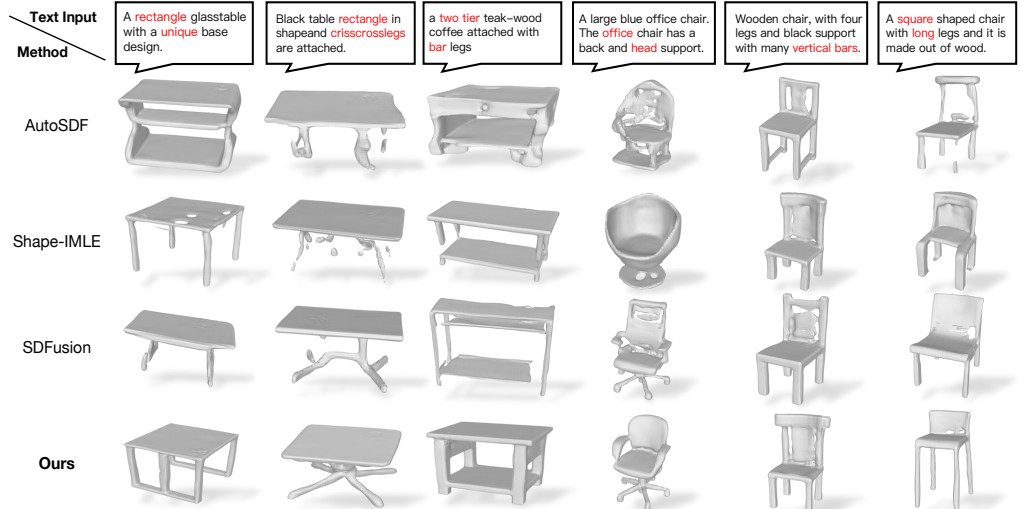

Figure 4: Visualization of the results of our method compared to AutoSDF, Shape-IMLE, and SD-Fusion. Our method generates satisfactory shapes that accurately align with multiple keywords (highlighted in red) from the input text description.

### 5.2 ENHANCED STRUCTURE-AWARE 3D SHAPE MODIFICATION

Existing approaches to text-guided 3D shape modification often fall short in achieving precise structural alignment. For example, when given a prompt "incline legs", these methods either struggle to produce effective, high-quality, and diverse shape modifications or generate ambiguous associations between the specified entity and its structural components (e.g., the adjacent seat). Our goal is to

Table 1: Quantitative generation results on random 1000 samples of Text2shape dataset.

| Method | IoU↑ | CLIP-S↑ | TMD↑ | EMD↓ |
|---|---|---|---|---|
| AutoSDF (Mittal et al., 2022) | 5.77 | 31.65 | 0.341 | 0.2659 |
| Shape-IMLE (Liu et al., 2022) | 12.21 | 31.42 | 0.672 | 0.2071 |
| SDFusion (Cheng et al., 2023) | 12.78 | 31.78 | 0.837 | 0.1792 |
| 3DQD (Li et al., 2023a) | 13.65 | 32.11 | 0.896 | 0.1767 |
| **Ours** | **13.87** | **32.65** | **0.910** | **0.1472** |

modify the input shape $X'$ to accurately reflect the text prompt $T'$, while preserving the integrity of unrelated regions. Our method, HierT2S, addresses this challenge by explicitly defining the text's structural elements early in the process. This enables localized and accurate shape modifications in line with the provided text prompt $T'$.

As illustrated in Figure 5, our method mainly follows the approach in (Couairon et al., 2022) to identify the region marked as $[MASK]$ $\Omega$ and modify. We enhance the input noise $M_T$ with two additional channels: one represents the $[MASK]$ region $\Omega$, and the other depicts the shape $\tilde{X}$ without the masked area. The additional channels are initialized with zero weights, while other model parameters are set using pre-trained weights. We proceed with fine-tuning the model for $t$ steps to adapt the masked region and generate the shape $\hat{X}$ that adheres to the prompt $T'$.

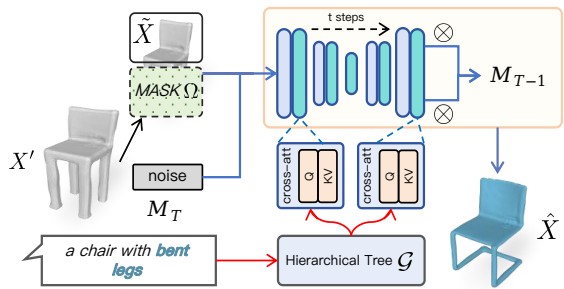

Figure 5: Pipeline for structure-aware 3D shape modification: After fine-tuning, our model performs localized modifications and generates coherent, text-aligned shapes.

In Figure 6, we demonstrate how the alignment capability of text descriptions enables precise and convenient shape manipulation. As shown in the figure, compared with 3DQD (Li et al., 2023a), HierT2S can effectively remove or add a specific part (such as the armrest of a chair) following the text instructions. We can easily modify part-level structures, such as transforming the straight legs of a chair into curved or angled ones, changing a single-layer table into a two-layer one, or even converting a square tabletop into a round one. Besides, as shown in Figure 7, our model allows for incremental modifications, while the impact on other unaffected regions remains minimal.

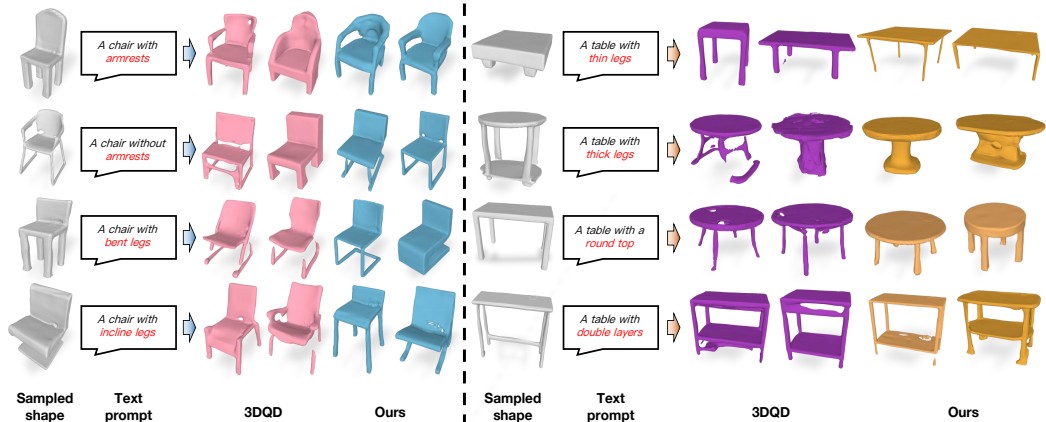

Figure 6: Qualitative results of text-guided shape manipulation compared with 3DQD. Given a known shape, our approach is able to manipulate the given shape into the target shape with prompt.

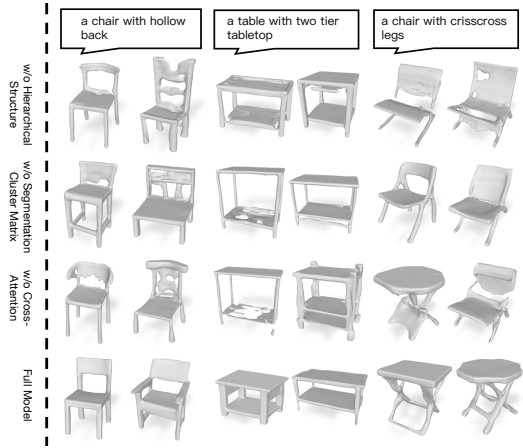

Figure 7: By incrementally adding prompts, our model enables high-quality modifications with minimal impact on unrelated regions.

## 5.3 ABLATION STUDY

We conducted ablation studies on the Text2Shape dataset to demonstrate the effectiveness of several key components of our method (Table 2). The variants tested were as follows: (1) **w/o Hierarchical Structure**: A sequential BERT-based text encoder was used, which does not incorporate hierarchical structural features; (2) **w/o Segmentation Cluster Matrix (SCM)**: Semantic features were directly integrated into the diffusion latent space via cross-attention in the Relation Graph Module, without employing SCM; (3) **w/o Cross-Attention**: Instead of using cross-attention, we concatenated the text features to the diffusion latent variables; (4) **Full Model**: Our complete method, including the hierarchical structure and all proposed components.

As shown in Figure 8, we visualize the results of each module in the process of semantic-guided 3D shape generation. It shows that the hierarchical relationship pretraining from text and learning embeddings contribute to generating detailed 3D parts from text, thus enhancing the performance of subsequent steps.

| Model | IoU↑ | CLIP-S↑ | TMD↑ | EMD↓ |
|---|---|---|---|---|
| w/o Hierarchical Structure | 11.68 | 31.92 | 0.891 | 0.1785 |
| w/o Segmentation Cluster Matrix | 13.04 | 32.03 | **0.924** | 0.1527 |
| w/o Cross-Attention | 11.24 | 30.75 | 0.847 | 0.1801 |
| Full Model | **13.87** | **32.65** | 0.910 | **0.1472** |

Table 2: Quantitative results of the ablation study for different model configurations.

Figure 8: The visualized results of ablation in three circumstances.

## 6 CONCLUSION

We presented HierT2S, a novel framework for text-to-shape generation and modification that exploits hierarchical structures inspired by human reasoning. The key contribution of this work is the use of a graph structure to impose a hierarchy on text, corresponding to the structure of 3D shapes and embedding relational features into a conditional diffusion model for structure-aware generation. Specifically, we employ the Hierarchical Tree to segment text into clusters and capture the relational embeddings of entities, which are then utilized in the conditional diffusion model to generate high-quality 3D shapes through joint training. Our approach surpasses existing methods in its ability to create structure-aware 3D shapes and facilitate precise, step-by-step shape manipulation using text. Extensive experiments demonstrate that our method improves generation quality and preserves the hierarchical characteristics of the shapes.

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
