# OpenReview forum: "HierT2S: Enhancing Part-Level Text-to-Shape Generation via Hierarchical Structure Modeling"
_ICLR.cc/2025/Conference — ICLR 2025 Conference Withdrawn Submission_

### Official Review · Reviewer_JM8g · 2024-10-31

**Soundness:** 3
**Presentation:** 3
**Contribution:** 3
**Rating:** 6
**Confidence:** 4

**Summary:**

HierT2S introduces a hierarchical structure modeling approach, where text is segmented into a tree-based structure, allowing for more precise alignment between text descriptions and 3D shape components. By utilizing a Relation Graph Module and a conditional diffusion model, the framework captures hierarchical relationships and generates structure-aware 3D shapes.

Key contributions include:
1. The authors parse text descriptions into a hierarchical tree structure, segmenting text into clusters representing different components. This tree captures the relationships within the text that correspond to part-level features in the 3D shapes.
2. Using a graph-based approach with self-attention mechanisms, this module aggregates relationships within text clusters, producing relational text embeddings that inform the 3D shape generation process.
3. HierT2S integrates the hierarchical text features into a conditional diffusion model, allowing for structure-aware 3D shape generation. This framework enhances fidelity to textual descriptions by accurately mapping part-based instructions to the generated shapes.
4. Beyond generation, the model supports progressive shape manipulation, allowing users to adjust individual shape parts in response to modified text prompts.

**Strengths:**

1. HierT2S segments input text into a hierarchical tree, which captures the structural relations in text descriptions and aligns them with the structure of 3D shapes. This method improves coherence and structural integrity in generated shapes by treating different components of the object separately.

2. Experiments demonstrate that HierT2S outperforms methods like AutoSDF, Shape-IMLE, and SDFusion, particularly in achieving high fidelity in part-specific shape details and maintaining consistency with text prompts. The paper also includes a detailed ablation study to illustrate the impact of each model component (e.g., hierarchical structure, segmentation cluster matrix, cross-attention). This systematic analysis reinforces the model’s design choices and provides clarity on the contribution of each module.

3. The model can manipulate specific parts of a shape according to new prompts, adding to its versatility for interactive applications.

4. The paper is well-structured and generally clear in its explanations. The methodology is broken down logically, with each component (e.g., hierarchical tree, relation graph module, conditional diffusion model) explained step-by-step. The figures, particularly Figures 1 and 2, visually support the hierarchical structure and relational embeddings used in the model, helping readers grasp the core ideas more intuitively.

**Weaknesses:**

1. The model sometimes produces minor artifacts, particularly in transitions between different parts of the generated shapes, which can slightly detract from overall visual coherence. While the hierarchical tree model effectively segments text into part-level components, there is limited discussion on the constraints and challenges associated with this representation. For instance, it is unclear how well this approach scales when dealing with more complex or ambiguous prompts, such as “a table with intricate detailing on the legs and an engraved top,” where elements might have multiple substructures or unclear hierarchical relationships.

2. The evaluation section lacks a detailed comparison with other methods specifically designed for part-level or fine-grained 3D shape generation. Although the model outperforms existing baselines on metrics like IoU and EMD, it is unclear how HierT2S compares with recent models that incorporate part-based structures or modular designs. Such comparisons would be particularly valuable given the paper’s focus on part-level control.

3. The paper currently lacks a comparison of training time with other state-of-the-art methods, making it difficult to evaluate the computational efficiency of this approach. Without insights into the time and resources required for training, readers may find it challenging to assess whether this method is feasible for deployment in environments with limited computational capacity.

**Questions:**

Could you elaborate on how the hierarchical tree handles complex prompts with nested or ambiguous descriptors (e.g., “a round table with intricate legs and a woven base”)? Are there specific techniques in place to manage such cases? It would be helpful to provide examples or guidance on how HierT2S processes more complex prompts and if alternative parsing techniques (e.g., dynamic clustering) could be considered to enhance model performance with nuanced descriptions.

---

### Official Review · Reviewer_oqJG · 2024-11-01

**Soundness:** 2
**Presentation:** 2
**Contribution:** 2
**Rating:** 3
**Confidence:** 5

**Summary:**

This paper presents a hierarchical structure modeling method to enhance part-level text-to-shape generation. The proposed method includes two phases, building the hierarchical tree and training the conditional diffusion model. The key idea is to learn the tree-like hierarchy of the text, through the implicit segment cluster and the relation graph module. For the implicit segment cluster, a segmentation cluster matrix is proposed to cluster nodes into groups. For the relation graph module, the attention mechanism is used to learn the relationship between cliques. In the experiments, the proposed method is evaluated on the public Text2Shape dataset, and compared with several existing methods.

**Strengths:**

Here I highlighted the strengths of the paper:
1. The problem this paper focuses on, the hierarchy of text and 3D shape, is worthy of study in the field of 3D content generation.
2. The proposed method outperforms the baseline method. The ablation study also shows the effectiveness of each module.
3. This paper is easy to read.

**Weaknesses:**

Although the problem this paper focuses on is valuable, this paper only proposes a simple method from the technical view and the innovation is limited. Here I highlight my major concerns.

1. The novelty of the proposed method is limited, that is the combination of existing methods. Firstly, the proposed method is similar to SDFusion except for the part of text feature learning. Second, when learning text features, the proposed segmentation cluster matrix does not substantially improve the Tree-transformer and the proposed relation graph module is just the use of the attention mechanism.

2. The evaluation of the effectiveness of hierarchical feature learning is insufficient. Though this paper compared with existing text-to-shape methods, such as SDFusion and 3DQD, which do not utilize hierarchical information, this paper didn’t cite and discuss existing hierarchical-based methods and didn’t provide a comparison with them. Comparing these methods can justify whether the proposed hierarchical feature learning is effective. These methods include [1, 2, 3 ....].
[1] ShapeScaffolder: Structure-Aware 3D Shape Generation from Text
[2] ShapeCrafter: A Recursive Text-Conditioned 3D Shape Generation Model
[3] HyperSDFusion: Bridging Hierarchical Structures in Language and Geometry for Enhanced 3D Text2Shape Generation

3. About long text descriptions. In the abstract, this paper points out that ignoring hierarchical modeling affects long text processing, but the experiment results on long text are missing in the experimental section.

4. About the visualization of the hierarchical tree. I recommend giving visual examples of hierarchical trees built, such as for a text, drawing its hierarchical tree. Otherwise, it is now difficult to know what the hierarchical structure of the learned text tree is.

5. About the definition of metrics. On page 8, the metrics, such as TMD and EMD, are missing definitions or references.

**Questions:**

As mentioned by weakness.

---

### Official Review · Reviewer_3YeG · 2024-11-01

**Soundness:** 2
**Presentation:** 3
**Contribution:** 1
**Rating:** 3
**Confidence:** 4

**Summary:**

The paper introduces a framework that enhances text-driven 3D shape generation by integrating a hierarchical tree representation with a conditional diffusion model. The method segments text into clusters, constructs a hierarchical tree, and processes these with a relation graph module to generate detailed 3D shapes. The technical contributions are minor. Authors failed to compare with the existing state-of-the-art 3D generation methods.

**Strengths:**

- Clear writing.

**Weaknesses:**

- While combining existing methods can sometimes lead to innovative solutions, the paper lacks a clear rationale for why these specific methods were chosen and how their integration results in a synergistic effect that advances the field. For instance, the text structures in ShapeNet dataset are typically simple, which can be regarded as the format of “{Adj.} {Noun.} … {Adj.} {Noun.}”. In this case, will the extracted text structures have big variances? Authors need to provide quantitative and qualitative analysis on the extracted trees. Moreover, lots of pretrained language models already can understand the language structures. Authors need to evaluate the performance comparison between using the latest pretrained language models and the proposed framework.

- In Relation Graph Module, why do you use BERT instead of other advanced pretrained langugae models as prompt features?

- It would be beneficial to assess whether the integration of the text parsing method, graph relation extraction, and 3D U-Net results in any novel architectural or methodological advancements. If the combination does not introduce new mechanisms or processes, it could be perceived as a straightforward aggregation of existing methods rather than a meaningful contribution. I suggest the authors explore any novel interactions or dependencies between the components that could lead to unique insights or improved performance metrics, and highlight these in the results.

- Authors only conduct experiments on Chairs and Tables of ShapeNet dataset. They generate 3D geometry only, without texture. This is too weak, given there already have works that can generate textured 3D objects in ShapeNet now, such as TAPS3D. In Objaverse dataset, there are lots of methods, such as Wonder3D, Unique3D and etc., that can generate much more complex 3D meshes. Authors need to provide comparison with these methods. Here are the suggestions:
1. **Broaden the Dataset Scope**: Encourage the authors to evaluate their method on the Objaverse dataset, which includes a diverse range of 3D objects.
2. **Texture Generation**: Suggest that the authors conduct experiments focusing on texture generation.
3. **Computational Efficiency**: Time taken for generation and resource utilization.

**Questions:**

Authors are advised to provide more justifications on why you chose certain methods as your model components, which should be improved in both writing and experiment parts. Please see my above comments in Weaknesses for details.

---

### Official Review · Reviewer_PHDb · 2024-11-02

**Soundness:** 2
**Presentation:** 2
**Contribution:** 1
**Rating:** 3
**Confidence:** 4

**Summary:**

This paper proposed HierT2S, a new framework that improves text-driven 3D shape generation by using hierarchical structures from both text and shapes. It breaks down text into clusters and builds a hierarchical tree to better understand part-level details. It then uses these structured text features in a diffusion model to create 3D shapes that are more semantically consistent and detailed. Tests on the Text2Shape dataset show HierT2S outperforms current methods, enabling more precise and progressive 3D shape edits based on text descriptions.

**Strengths:**

- I believe the studied direction -- from part-level understanding for shape generation is important to our community.
- The proposed framework are designed in a reasonable way, including using relation graph module and hierarchical tree stuffs.
- They compared with some well-known baselines, including AutoSDF, ShapeIMLE, and SDFusion. Also, the paper provides some ablation studies.

**Weaknesses:**

- The results shown in Figure 1 & 4 are low-quality as the geometry details are fuzzy. Even consider geometry solely without considering many other frameworks do produce colors, the results are not good enough.
- Some of the design choice is not good enough. For example, the sampling point clouds only have 2,048 points which could be the reason the results cannot recover the back strip for the chair. Can the author try higher point samplings, like 8,192?
- I am confused about how the proposed Hierarchical Structure works. Does the learned full node connection really make sense? Can the author provide more in-depth analysis here instead of just final results along with some inputs. I failed to understand how the parts affect the final generation performance. Can we visualize the learned node connections?
- The current paper only performed on very old data Text2Shape as well as ShapeNet. However, we have much more 3D assets such as ABO and Objaverse. Results demonstrate on those newer datasets could be appealing. The current tested text-prompt is really simple.

**Questions:**

- please address the weakness raised above.
- Can the author easily change the framework to adopt colors instead of geometry solely?

---

### Official Review · Reviewer_j3Ay · 2024-11-03

**Soundness:** 3
**Presentation:** 3
**Contribution:** 3
**Rating:** 6
**Confidence:** 3

**Summary:**

Current text-to-shape models struggle with generating complex shapes with detailed descriptions, especially as they pertain to parts. This paper introduces a way to do this  by generating a hierarchical tree representation of text, paired with a conditional diffusion model, to better capture the relationship between text and the 3D shape parts. Experiments show that HierT2S is able to preserve structural consistency and support progressive shape manipulation.

**Strengths:**

The method seems to be well motivated, and the hierarchical structure modeling method seems to align the structure of language and 3D parts. The method enables part-level shpae manipulation, whcih allows for fine-grained control over shape edits (although I have some concerns outlined in the weaknesses section). The paper demonstrates strong empirical results on the Text2Shape dataset, and can be extended to other larger existing datasets, like ShapeTalk.

**Weaknesses:**

The method does not seem to produce part-disentangled edits -- For instance, in Figure 6, adding armrest to the chair changes the thickness of the legs, the width and the shape of the back as well.

Some prior work in this area have shown that part-disentanglement can naturally emerge from training with natural language (https://arxiv.org/abs/2112.06390 and https://arxiv.org/abs/2212.05011). Are any of the regularizations/tricks applied in these papers applicable to improving the part-disentanglement of the edits in your method?

**Questions:**

Could the method be used with contrastive utterances used to discriminate between two shapes? If so, it also means that you can leverage a much larger set of utterance data existing in larger datasets like the one
used in https://changeit3d.github.io/.

---

### Official Review · Reviewer_JWES · 2024-11-03

**Soundness:** 3
**Presentation:** 2
**Contribution:** 3
**Rating:** 5
**Confidence:** 4

**Summary:**

The paper presents HierT2S, a novel approach to part-level, text-to-3D shape generation that leverages hierarchical structure modeling. Traditional text-to-3D generation methods often treat shapes as single entities, overlooking hierarchical details that may be specified in complex text prompts. HierT2S addresses this by constructing a hierarchical tree from text, clustering descriptors that correspond to part-level structures (e.g., “seat” and “legs” for a chair) and feeding these relational embeddings into a conditional diffusion model. This method not only enhances the coherence of generated shapes but also enables part-level manipulation of 3D shapes. Experimental results on the Text2Shape dataset demonstrate that HierT2S outperforms state-of-the-art methods in generating detailed, structurally accurate 3D shapes.

**Strengths:**

1. I always like the idea of decomposing detailed input text as hierarchical structures. This paper further separates the generation of each part as a course-to-fine condition for the diffusion model.

2. HierT2S allows users to adjust specific parts of a 3D shape through updated text prompts, achieving localized shape modifications. This practical application of structure-aware manipulation at the part level builds on the relational focus of StructureNet but offers improved control, particularly in the context of text-driven, part-level adjustments.

3. The method demonstrates consistently high quantitative results in metrics like IoU, CLIP-S, TMD, and EMD, indicating effective semantic alignment and structural coherence. This reflects the model's capacity to balance part-level fidelity with generation quality.

**Weaknesses:**

1. Although I said I like the idea, much of the claimed novelty in HierT2S, specifically the hierarchical modeling of parts in 3D objects, is conceptually similar to prior works like StructureNet, PartNet, and Hierarchical Surface Prediction. This reduces the originality of HierT2S, positioning it as an incremental advance that mainly adds diffusion-based modeling to a well-explored framework. I expect to see different designs for the diffusion process that fit more into the idea of the hierarchical graph instead of just treating those decomposed information as conditions of different scales.

2. The model's hierarchical parsing approach assumes well-organized, descriptive text inputs, which may limit its robustness with less structured or ambiguous text. This makes it potentially less adaptable to a variety of real-world input scenarios compared to more flexible models that handle less structured data.

3. Evaluated solely on the Text2Shape dataset, HierT2S may lack robustness across diverse or complex categories, especially as hierarchical part relationships become more intricate. However, I understand that the paired text-3D dataset with detailed description is very limited. In that case, I suggest the authors incorporate the conditions of different models in this framework (e.g., combining shape completion, image-to-3D, and text-to-3D) to show more capability of this work. Current experiments and demonstrations are too narrow.

**Questions:**

1. What would happen if using a 2D captioning model to generate pseudo captions for images rendered from 3D models as training data of the text input, just like what TAPS3D (CVPR 2023) and TPA3D (ECCV 2024)? Does that help enlarge the capability of HierT2S, or does your text parsing network fail due to over-noisy text descriptions?

2. Following the last question, if your network fails to face the above situation, how can we apply it to real-world applications if there's only Text2Shape suits your method? If not, why are only experiments of Text2Shape shown in this paper while you can generate pseudo captions for training on other classes of objects?

3. How does your model deal with ambiguous descriptions, noisy descriptions, or over-complex ones? Can you provide some examples?

4. What's the limitations of this work?

---

### Note · Authors · 2024-11-13

I have read and agree with the venue's withdrawal policy on behalf of myself and my co-authors.